# Adverse Events Associated with Prolonged Antibiotic Therapy for Periprosthetic Joint Infections—A Prospective Study with a Special Focus on Rifampin

**DOI:** 10.3390/antibiotics12111560

**Published:** 2023-10-24

**Authors:** Pia Reinecke, Paula Morovic, Marcel Niemann, Nora Renz, Carsten Perka, Andrej Trampuz, Sebastian Meller

**Affiliations:** Charité—Universitätsmedizin Berlin, Corporate Member of Freie Universität Berlin, Humboldt-Universität zu Berlin and Berlin Institute of Health, Center for Musculoskeletal Surgery (CMSC), Augustenburger Platz 1, 13353 Berlin, Germany; pia.reinecke@charite.de (P.R.); paula.morovic@charite.de (P.M.); marcel.niemann@charite.de (M.N.); nora.renz@charite.de (N.R.); carsten.perka@charite.de (C.P.); andrej.trampuz@charite.de (A.T.)

**Keywords:** periprosthetic joint infection, arthroplasty, antibiotics, long-term antibiotic therapy, adverse events, rifampin

## Abstract

Periprosthetic Joint Infection (PJI) is a significant contributor to patient morbidity and mortality, and it can be addressed through a range of surgical interventions coupled with antibiotic therapies. Following surgical intervention(s), prolonged administration of oral antibiotics is recommended to cure PJI. There is a lack of reports on the adverse events (AEs) associated with oral antibiotics, particularly rifampin. This investigation sought to elucidate the occurrence of antibiotic-related AEs after an initial regimen of intravenous antibiotic administration, supplemented by an extended course of oral antibiotics. A prospective study of patients diagnosed with PJI of the hip, knee, or shoulder who underwent single-stage exchange arthroplasty (SSE) (10%), two-stage exchange arthroplasty (TSE) (81%), or debridement, antibiotics, and implant retention (DAIR) (6%) was performed. The primary outcome of interest was the detection of AEs, the secondary outcome the detection of a correlation between rifampin use and the incidence of AEs, and the tertiary outcome was whether oral antibiotic treatment needed to be adjusted or discontinued due to AEs. In addition, subjective tolerability was monitored throughout the study. A total of 336 events were identified for 73 out of 80 patients. The most frequently used antibiotics were rifampin and co-trimoxazole. Most AEs occurred in the gastrointestinal tract (46%). The most frequent AEs were nausea, inappetence, diarrhea, and skin rash. In 6% of cases, the AEs led to antibiotic discontinuation, and in 29% of cases, a dose adjustment of the oral therapy occurred, mainly with amoxicillin or co-trimoxazole. The majority of patients (55%) rated the subjective tolerability as good. In conclusion, AEs during antibiotic treatment for PJI are common. They mainly affect the gastrointestinal tract. Rifampin use might be a reason for the higher incidence of AEs compared to non-rifampin antibiotic treatment.

## 1. Introduction

Periprosthetic joint infection (PJI) is a severe complication in orthopedic surgery [1]. Infections are one of the most common causes of failure of revision arthroplasty [2,3,4]. Colonization of the prosthesis leads to biofilm formation, which is one of the main microbiological factors responsible for the observed outcomes of PJI and orthopedic device-related infections. Furthermore, the treatment of PJI is accompanied by increased psychosocial stress for patients comparable to that of oncology patients [5]. Bernard et al. recently showed that nonserious adverse events (AEs), mainly gastrointestinal disorders and mycosis, were more common in patients treated with a 12-week course of antibiotics than in those treated with a 6-week course [6]. Long-term antibiotic therapy is part of the standard PJI management and is adapted to the susceptibility of the pathogen(s), patient characteristics (weight, renal/liver function, allergies), and the performed surgical strategy [7].

The topics relevance results from the expected rise of PJI cases in the future [8,9]. This is due to an aging society and the resulting increase in demand for joint replacement surgery to maintain mobility and independence, as well as the longer indwelling time of prostheses. 

The efficacy of rifampin in treating staphylococcal PJI was shown in vitro, in experimental animal models, and in several clinical studies [10,11,12]. Therefore, rifampin is included in most national and international guidelines; however, AEs and interactions with other drugs make its use challenging [10,13]. There is limited data on the frequency, type and severity of AEs following long-term antibiotic treatment for PJI. In particular, data on the subjective tolerability of antibiotic treatment for PJI, as reported by patients, are lacking. 

We assessed patients’ tolerability of oral antimicrobial treatment in an institutional PJI cohort. The study focuses on the frequency and type of specific AE, and the influence of adding rifampin on their occurrence and tolerability.

## 2. Results

### 2.1. Patients

During the study period, 126 patients were treated for PJI in our institution. After excluding 46 patients (5 patients due to a language barrier, 11 due to non-PJI-related death, 17 due to loss to follow-up, and 13 due to incomplete medical records), 80 patients were included in the analysis. The median patient age was 69 years (range 27–92 years), 39 (48%) were female. Most PJI involved hip prostheses (85%, *n* = 68), followed by knee (13%, *n* = 10) and shoulder prostheses (3%, *n* = 2). Two-stage exchange arthroplasty (TSE) was performed in 65 patients (81%) and single-stage exchange arthroplasty (SSE) in 8 patients (10%). Debridement, antibiotics and implant retention (DAIR) was performed in 5 patients (6%), and 2 patients remained prosthesis-free after explanation of the implant.

### 2.2. Antibiotic Treatment

Patients received antibiotic treatment for a median of 12 weeks (range 4–168 weeks). The most commonly used antibiotic was rifampin (68%, *n* = 54), followed by sulfamethoxazole/trimethoprim (co-trimoxazole) (53%, *n* = 42). The daily dose of rifampin was 900 mg (72%; *n* = 39) or 600 mg (28%; *n* = 15) given in 1–2 doses. The proportions of patients per antibiotic group are shown in Table 1.

### 2.3. Adverse Events

The median duration of follow-up for the patients enrolled was 28 months (range 2–47 months), starting from the date of the last surgery. Overall, 336 AEs occurred in 73 patients. Seven patients (9%) reported no AEs. A total of 48 patients (60%) had 5 or fewer AEs, 19 patients (24%) had more than 5 AEs, and six patients (8%) had 10 or more AEs during the antibiotic treatment. Most of the AEs affected the gastrointestinal tract (156/336; 46%), followed by the skin and skin appendages (74/336; 22%), and the peripheral or central nervous system (46/336; 14%). The most frequently occurring AEs were nausea, diarrhea, inappetence, skin rash, dry skin, and hair or nail changes (Table 2).

Twenty-three patients (29%) needed an adjustment of their therapy due to AEs, which included a reduction of the dosage or switch to another substance. The therapy was discontinued earlier than planned in five cases (6%). Adjustment and discontinuation were primarily carried out in patients receiving co-trimoxazole (*n* = 12) and amoxicillin (*n* = 10). The median time to adjustment or discontinuation was seven days (range 1–28 days). The AEs that led to a therapy switch or discontinuation of the antimicrobial substance are listed in Table 3. No patient reported cholestasis, jaundice, or tendinopathy.

Patients receiving antibiotic treatment including rifampin had a higher incidence of AEs than patients not receiving rifampin (240/336 (71%) vs. 95/336 (28%), *p* = 0.399). There were no differences in the incidence of AEs in patients stratified according to sex or age of the patients (Table 4).

### 2.4. Subjective Tolerability of Oral Antibiotic Therapy

Personal responses on the subjective tolerability of the antibiotic treatment were available from 76 of the 80 patients (41 male, 35 female). Subjective tolerability of the patients was rated as “good” in 55% of cases (*n* = 44), which was reported by more males (*n* = 29) than females (*n* = 15). A total of 25 patients (10 male and 15 female) rated the treatment tolerability as “mediocre” (31%), and 7 patients (2 male and 5 female) categorized their tolerance as “bad” (9%). There was a trend towards higher subjective tolerability of antimicrobial treatment in male participants compared to female patients (*p* = 0.02). Through direct patient interviews, it was ascertained that patients were cognizant of the potential emergence of AEs during their therapeutic course. Furthermore, they consistently reported that these adverse events primarily persisted for the duration of the antibiotic regimen and subsided rapidly upon cessation of the therapy.

## 3. Materials and Methods

### 3.1. Study Design 

After obtaining institutional board approval (EA2/059/20), we conducted a prospective study of patients who underwent revision surgery for hip, knee, or shoulder PJI in our specialized septic surgery department at a single academic institution between January 2020 and December 2022. The diagnosis of PJI was based on institutional definition criteria, and the management of PJI was performed by an interdisciplinary team specialized in musculoskeletal septic surgery [14,15,16]. Included were acute and chronic PJI of hip, knee, or shoulder arthroplasties that were surgically treated with SSE, TSE, or DAIR. All patients received a course of antimicrobial treatment for at least six weeks, according to our protocol [16].

The exclusion criteria were: (1) patients under the age of 18 years, (2) patients who dropped out of the study due to a language barrier, loss to follow-up, patients with an incomplete medical record, or (3) patients who died before the reimplantation surgery due to reasons not related to PJI.

### 3.2. Definitions of AEs

The criteria used to define the AEs are summarized in Table 5. These definitions were derived from previous publications, institutional standards, and consensus opinions. 

### 3.3. Data Collection

We prospectively collected data from personal interviews with the patients by telephone or clinical visits. We used a standardized questionnaire listing AEs (shown in Table 2) described in previous studies [14,15,16,17,18,19,20,21,22,23,24,25,26,27] and documented in our clinical reports. Additional AEs were noted upon being mentioned. To measure patients’ subjective perceptions, we asked them to rate the tolerability of the antibiotic treatment on a scale from 0 to 10 (0 = bad, 10 = good). In the next step, we summarized the results into three categories. “good” (8–10), “mediocre” (4–7.5), and “bad” (0–3.5). Patients were divided into two groups based on rifampin use. Rifampin was always administered in combination with another antibiotic.

### 3.4. Treatment

We performed a DAIR procedure for acute infections in patients with well-fixed components and in late acute PJI. For chronic PJI with symptoms lasting more than four weeks, we treated patients with SSE or TSE, according to the institutional treatment protocol [17], followed by initial intravenous antibiotic therapy for a maximum of 14 days. According to the protocol, the standard antibiotic regimen, including intravenous and oral treatment, consisted of a 12 week-course of targeted antimicrobial therapy. In this analysis, the focus was on AEs experienced during oral antibiotic treatment only. The standard rifampin dose was 450 mg twice daily. In patients over 75 years of age with a body weight of less than 60 kg, or who developed AEs related to the full dose of rifampin, the dose was reduced to 300 mg twice daily or 600 mg once daily. In cases where treatment had to be changed due to side effects, either the dosage of the causative antibiotic was reduced (after a pause of two to three days), it was replaced by an equivalent alternative preparation, or the treatment was discontinued completely.

### 3.5. Statistical Analyses

Descriptive statistics were performed in all outcome measures. The median and ranges were used for continuous variables. Categorical variables are shown using counts and percentages. Differences between groups were assessed using the Mann–Whitney U test for continuous variables and Fisher’s Exact Test for categorical variables. In addition, *p*-values < 0.05 were considered significant. IBM^®^ SPSS^®^ Statistics (Version 27) was used for all statistical analyses.

## 4. Discussion

Our study indicates that AEs during antibiotic treatment for PJI are common, mainly involving the gastrointestinal tract. The concern of PJI treatment failure was high [18]. However, although almost all patients experienced at least one antibiotic-related AE, long-term antibiotic therapy was completed by the majority of patients. Only five patients required treatment discontinuation, suggesting that long-term antibiotic treatment is commonly accepted despite the many AEs. To our knowledge, this is the first prospective study to present all AEs that occurred during long-term antibiotic therapy in PJI treatment in detail and demonstrate the subjective tolerability of the antibiotic treatment.

Comparisons with other studies that have used antibiotic regimens for PJI for at least six weeks is difficult because others have focused primarily on evaluating the efficacy of specific antibiotic treatment protocols, not treatment tolerability and adherence. For example, Schindler et al. [19] analyzed AEs when treating osteoarticular infections in general, not specifically PJI, and included oral and intravenous therapy. The overall incidence of AEs in previous studies varies from 14 to 53% [13,19,20,21,22,23,24,25,26,27,28,29,30,31]. Focusing on the proportion of AEs per organ system makes our data comparable with other studies. The gastrointestinal tract was the most affected organ system, followed by the skin and skin appendices [13,19,20,21,22,23,24,25,26,27,28,29,30,31]. Hematologic AEs were reported as being uncommon, possibly because they were only recorded when they were severe or occurred during a hospital stay. Oral antibiotics were administered in an outpatient setting, which was typically attended to by a general physician, not the hospital specialists, and were, therefore, rarely recorded in the hospital patient record.

Nevertheless, according to the rates of AEs, our results seem slightly higher than those of others. One reason for this might be that we asked for any AEs that the patients might overlook. Patients were also more likely to remember AEs they had seen or experienced daily, such as a severe rash or diarrhea. Hence, these events were increasingly reported. Other events, such as anemia or ECG changes, are not directly noticeable to the patients, are not as limiting, and may not be remembered.

In other studies, the rate of treatment adjustment was lower [13,19,20,21,22,23,24,25,26,27,28,29,30,31] compared to our observations. The discontinuation rate was comparable, 6% versus 4% [13,19,20,21,22,23,24,25,26,27,28,29,30,31]. As a specialized center [16], we are more likely to deal with patients who already have several comorbidities and are treated with antibiotics. Therefore, treatment adjustments due to resistance, drug interactions, or AEs might be more frequent than in other medical centers. We try to avoid discontinuing antibiotic therapy in all cases where adjustment is possible. The antimicrobial treatment duration ranged from 4 to 168 weeks in our study. The treatment duration of several years is explained by suppressive antimicrobial treatment administered to patients suffering from infections caused by microorganisms for which no biofilm-active agent was available, or in patients where the performed treatment strategy was not appropriate to eradicate the infection (e.g., retention of the prosthesis in the case of chronic infection).

Schindler et al. [19] concluded that rifampin might be preventive against Clostridioides difficile enterocolitis, which may be an explanation for the low incidence in our study. Only three of the 54 patients receiving rifampin developed Clostridioides difficile infection.

In our study, patients receiving rifampin had a higher incidence of AEs than those not receiving rifampin (71% vs. 28%). Nguyen et al. [13] proved that high dosages of rifampin (>600 mg) led to more AEs, but did not result in a better outcome than therapy with a lower dosage. In our study, more patients (72%; *n* = 39) received higher dosages of rifampin (900 mg per day). This could serve as a potential explanation for the increased incidence of AEs. Although the negative effects of rifampin are known, such as a higher incidence of AEs in this case, this fact may be outweighed by the beneficial characteristics of rifampin, including its excellent oral bioavailability, high degree of effectiveness against *Staphylococcus* infection, and high biofilm activity [12,32]. Furthermore, rifampin was the reason for discontinuation or adjustment in only 2 cases out of 28. We assume that the increased incidence of AEs with rifampin treatment may be due to its unique efficacy against biofilms, leaving us with limited antibiotic alternatives. In contrast, co-trimoxazole demonstrated greater flexibility in dose adjustment, which may have contributed to its comparatively lower incidence of AEs. The fact that AEs occurred less in patients receiving non-rifampin therapy must be interpreted carefully.

As rifampin is consistently administered as part of a combination therapy, establishing its definite role as the primary causative antibiotic in AEs is challenging. However, considering that monotherapy with rifampin results in an increased risk of the development of resistance, and contradicts the current literature [33,34], its association with AEs remains unknown.

In the assessment of patients’ subjective tolerability, the majority receiving treatment classified the antibiotic therapy as either good or mediocre and indicated a predisposition to receiving similar treatment in subsequent instances. These observations suggest that patients might manifest a positive subjective tolerance towards extended antibiotic therapy and demonstrate adherence to the prescribed therapeutic protocol. Notably, there appears to be a tentative trend suggesting that male patients might demonstrate enhanced tolerance to the treatment relative to their female counterparts. However, it is important to emphasize that, to our knowledge, there is no further literature on this.

As a single-center study with a limited number of patients, our results should be interpreted in the context of some limitations. In some cases, it was not possible to precisely link an AE to antibiotic treatment. Therefore, it is difficult to say which therapy causes more AEs. Among the 28 patients requiring treatment adjustments, the precise causation was attributed based on comprehensive documentation within their medical records. Most patients had comorbidities and had considerable co-medication. The described tolerability of the patients was based on subjective interviews.

## 5. Conclusions

Comprehensive patient briefing regarding potential AEs is crucial before commencing extended antibiotic treatments, priming them for the journey ahead. Given the inevitability of AEs, efforts must pivot towards their mitigation. Validating the hypothesis that reduced rifampin dosages yield better tolerability and fewer AEs necessitates a broader study with a larger sample size, especially when determining the correct rifampin dose regime for PJI therapy.

## Figures and Tables

**Table 1 antibiotics-12-01560-t001:** Number and proportion of patients categorized in antibiotic groups.

Rifampin Combination Therapy, *n* = 54 (68%)	Non-Rifampin Based Therapy ^1^, *n* = 26 (33%)
Rifampin + levofloxacin (33, 61%)	Amoxicillin (11, 42%)
Rifampin + co-trimoxazole (8, 15%)	Co-trimoxazole (10, 39%)
Rifampin + ciprofloxacin (5, 9%)	Ciprofloxacin (10, 39%)
Rifampin + linezolid (4, 7%)	Doxycycline (6, 23%)
Rifampin + doxycycline (4, 7%)	Clindamycin (5, 19%)
	Linezolid (4, 15%)
	Levofloxacin (1, 4%)

Data are shown as no. of patients (%). ^1^ Some antibiotics were used for combination therapy; therefore, patients were sorted into several groups.

**Table 2 antibiotics-12-01560-t002:** Frequency of AEs stratified by the involved organ system affected.

Organ System	*n* (%)	Adverse Events	*n* (%)
Gastrointestinal tract	156 (46)	Nausea	28 (35)
		Diarrhea	27 (34)
		Inappetence	22 (28)
		Reflux	15 (19)
		Dry mouth	15 (19)
		Vomiting	13 (16)
		Dysgeusia	10 (13)
		Constipation	6 (8)
		Meteorism	6 (8)
		Clostridioides difficile enterocolitis	6 (8)
		Malaise	4 (5)
		Change in color or texture of the tongue	4 (5)
Skin and skin appendages	74 (22)	Skin rash	20 (25)
		Dry skin	19 (24)
		Pruritus	16 (20)
		Candida mucositis (oral and genital)	12 (15)
		Changes in hair or nails	3 (4)
		Skin swelling	2 (3)
		Photosensitivity	2 (3)
Bone marrow/blood	9 (3)	Anemia	5 (6)
		Leukopenia	2 (3)
		Thrombocytopenia	1 (1)
		Pancytopenia	1 (1)
Liver	5 (1)	Hepatitis	5 (6)
Kidney	14 (4)	Renal insufficiency	14 (18)
Peripheral and central nervous system	46 (14)	Fatigue	11 (14)
		Vertigo	8 (10)
		Visual problems	6 (8)
		Headache	5 (6)
		Memory issues	5 (6)
		Neuropathy, paresthesia	4 (5)
		Symptoms of depression	4 (5)
		Anxiety	2 (3)
		Dizziness	1 (1)
Other	32 (10)	Weight loss	16 (20)
		Weight gain	5 (6)
		Dyspnea	5 (6)
		Myalgia and/or arthralgia	4 (5)
		Changes in electrocardiogram (long QT)	2 (3)

**Table 3 antibiotics-12-01560-t003:** List of 28 patients with adverse events requiring treatment adjustment of causative antimicrobial agent.

Causative Antibiotic	Patient	Adverse Event	Adjustment
Amoxicillin and clavulanic acid	1	Nausea and vomiting	Discontinuation
	2	Nausea, vomiting, skin rash	Change of antibiotic
Amoxicillin	3	Dysgeusia	Dose reduction
	4	Chronic diarrhea	Discontinuation
	5	Diarrhea	Dose reduction
	6	Skin rash	Change of antibiotic
	7	Skin rash, pruritus	Change of antibiotic
	8	Allergic reaction mainly of the skin	Dose reduction
	9	Not defined intolerance	Dose reduction
Amoxicillin and co-trimoxazole	10	Nausea and inappetence	Change of antibiotic
	11	Skin rash, renal insufficiency	Discontinuation of amoxicillin and dose reduction of co-trimoxazole
Clindamycin and co-trimoxazole	12	Clostridioides difficile, renal insufficiency	Discontinuation of clindamycin and dose reduction of co-trimoxazole
Co-trimoxazole	13	Skin rash	Change of antibiotic
	14	Skin rash	Change of antibiotic
	15	Skin rash and pruritus	Change of antibiotic
	16	Leukopenia	Change of antibiotic
	17	Renal insufficiency	Discontinuation
	18	Renal insufficiency	Discontinuation
	19	Renal insufficiency	Change of antibiotic
Co-trimoxazole and ciprofloxacin	20	Nausea, inappetence; oral candida infection	Dose reduction of co-trimoxazole, discontinuation of ciprofloxacin
	21	Skin rash; delirium and speech disorder	Change of both antibiotics
Levofloxacin	22	Color change of tongue	Change of antibiotic
	23	Oral candida infection	Change of antibiotic
Moxifloxacin	24	Nausea and not defined intolerance	Change of antibiotic
Rifampin	25	Hepatitis and cardiac decompensation	Discontinuation
	26	Leukopenia	Change of antibiotic
Linezolid	27	Pancytopenia	Change of antibiotic
Doxycycline	28	Skin rash and not defined intolerance	Change of antibiotic

**Table 4 antibiotics-12-01560-t004:** Comparison of frequency of AEs with regards to sex, age, and addition of rifampin.

Adverse Events According to Organ system ^1^	Female (*n* = 39)	Male (*n* = 41)	*p*- Value	Age < 65 (*n* = 24)	Age ≥ 65 (*n* = 56)	*p*- Value	Rifampin Combination Therapy (*n* = 54)	Non- Rifampin Therapy (*n* = 26)	*p*- Value
Gastrointestinal tract	28 (72)	30 (73)	0.890	18 (75)	40 (71)	0.743	40 (74.1)	18 (69.2)	0.65
Skin and Skin appendages	22 (56)	20 (49)	0.495	13 (54)	29 (52)	0.845	29 (54)	13 (50)	0.756
Bone marrow/blood	5 (13)	3 (7)	0.476	3 (13)	5 (9)	0.691	6 (11)	2 (8)	1.000
Liver	3 (8)	2 (5)	0.671	2 (8)	3 (5)	0.633	3 (6)	2 (8)	0.658
Kidney	5 (13)	9 (23)	0.28	4 (17)	15 (27)	0.333	14 (26)	5 (19)	0.585
Peripheral or central nervous system	12 (31)	14 (34)	0.747	7 (29)	19 (34)	0.677	17 (32)	9 (35)	0.779
Other ^1^	14 (36)	10 (24)	0.262	5 (21)	19 (34)	0.241	15 (28)	9 (35)	0.532
All adverse events	159/336 (47)	176/336 (53)	0.965	101/336 (30)	234/336 (70)	0.920	240/336 (72)	95/336 (28)	0.399

^1^ Patients had at least one adverse event in this category. Data are shown as no. of patients (%), except for the last line (no. of AEs (%)).

**Table 5 antibiotics-12-01560-t005:** Definitions of adverse events.

Adverse Event	Definition
Clostridioides difficile infection	Clinical signs and symptoms consistent with Clostridioides difficile infection in the setting of a positive Clostridioides difficile PCR test result
Anemia	Hemoglobin level < 10 g/dL
Leukopenia	White blood cell count < 4500 leukocytes/µL
Thrombocytopenia	Platelet count < 150.000/µL
Renal insufficiency	Increase in serum creatinine level > 1.5× patients’ baseline
Hepatitis	Aspartate transaminase or alanine transaminase level > 3× patients’ baseline
Changes in weight	Patients’ subjective impression of losing or gaining weight during the time of antibiotic treatment
Changes in electrocardiogram	Prolonged QT-time; >550 ms

## Data Availability

The data presented in this study are available on request from the corresponding author. The data are not publicly available due to local institutional ethics board regulations.

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
