# Peer review of "Adverse Events Associated with Prolonged Antibiotic Therapy for Periprosthetic Joint Infections—A Prospective Study with a Special Focus on Rifampin"

_antibiotics, 2023, doi:10.3390/antibiotics12111560_

Round 1
Reviewer 1 Report
This is a prospective cohort study that sought to elucidate the occurrence of antibiotic-related AEs focusing on oral antibiotics, particularly rifampin, administered as a sequencing regimen after 14 days of IV treatment in prosthetic joint infection (PJI). As notable data, the fact that 73/80 (91%) of the included patients presented some adverse effect, demonstrates the magnitude of the problem that physicians who treat infections that require antibiotic treatments for a prolonged period face. The originality of the work lies in the fact that although these AEs are usually observed by the physicians who treat these patients, until now no effort has been made to collect them prospectively and analyze them.
The work is well written, the methodology is adequate, and it may help to prevent and better mitigate these side effects, thus avoiding the suspension or changes of antibiotic treatments to alternative regimens.
However, I would suggest to the authors to consider the following comments.
1. Table 2 includes AEs Urinay tract infections and asymptomatic bacteriuria as AEs Does the author believe that these effects are associated with antibiotic treatment? If so, could you provide a possible explanation?
2. I suggest the authors merge tables 2 and 4. This way we could know which antibiotic each adverse effect is associated with, and the adjustment made. This would certainly be of great help to readers in their clinical practice.
3. I also suggest reviewing Table 4 and grouping the adverse effects that are similar (for example nephrotoxicity, acute renal failure and renal failure, skin rash (appears several times), allergic reaction of the skin and pruritus, oral candida and candida mucositis) associating them in column 2 with all the antibiotics that can cause it. In this way, we make it easier its revision for the reader
4. It would be interesting for the authors to explain if they have adopted any specific attitude to minimize any detected adverse effect and its result (for example: associating folic acid with treatment with cotrimoxazole to reduce myelotoxicity, monitoring levels of linezolid and adjusting doses or prescribing symptomatic treatment. to treat gastrointestinal effects such as nausea)
5. The authors could comment in the discussion on some less frequently observed effects such as the change in color of the tongue, which, unlike what the authors describe, in our experience, we see associated with treatments with amoxicillin and not with levofloxacin.
I hope these comments are useful and contribute to improving the quality of this work
Author Response
Point-by-point reply
Many thanks for reviewing and commenting on our manuscript which considerably improved our manuscript.
Comments of reviewer 1:
- Table 2 includes AEs Urinay tract infections and asymptomatic bacteriuria as AEs. Does the author believe that these effects are associated with antibiotic treatment? If so, could you provide a possible explanation?
Authors’ reply: We completely agree, this is not associated with antimicrobial treatment. We excluded “urinary tract infection” and “asymptomatic bacteriuria” from the tables and statistics. Accordingly, we also corrected the total number of AEs.
- I suggest the authors merge tables 2 and 4. This way we could know which antibiotic each adverse effect is associated with, and the adjustment made. This would certainly be of great help to readers in their clinical practice.
- I also suggest reviewing Table 4 and grouping the adverse effects that are similar (for example nephrotoxicity, acute renal failure and renal failure, skin rash (appears several times), allergic reaction of the skin and pruritus, oral candida and candida mucositis) associating them in column 2 with all the antibiotics that can cause it. In this way, we make it easier its revision for the reader
Authors’ reply: Thank you for this remark. In the previous version, it was probably not clear, that table 2 shows the total frequency of specific AEs in the cohort sorted by the organ system affected. In contrast, table 4 lists the AEs experienced by the 28 patients with the respective causative antimicrobia agent and the adjustment needed. The latter is of relevance for clinicians.
We adapted both tables graphically and contentwise to make them clearer and easier to understand for the reader.
- It would be interesting for the authors to explain if they have adopted any specific attitude to minimize any detected adverse effect and its result (for example: associating folic acid with treatment with cotrimoxazole to reduce myelotoxicity, monitoring levels of linezolid and adjusting doses or prescribing symptomatic treatment. to treat gastrointestinal effects such as nausea)
Authors’ reply: We added the following sentence to the methods section: “In cases where treatment had to be changed due to side effects, either the dosage of the causative antibiotic was reduced (after a pause of two to three days), it was replaced by an equivalent alternative preparation or the treatment was discontinued completely.”
We refrain from adding folic acid to cotrimoxazole treatment (as it acts as an antagonist) nor measure any drug levels. In case of mild nausea or vomiting, an antiemetic agent was administered 30min before the intake of the antimicrobial substance(s).
- The authors could comment in the discussion on some less frequently observed effects such as the change in color of the tongue, which, unlike what the authors describe, in our experience, we see associated with treatments with amoxicillin and not with levofloxacin.
Answer: The change in tongue color described in Table 4 occurred in only one patient out of four and was the only one requiring a change of antibiotic due to this AE. In the other patients, it was not possible to assign this AE to a specific antibiotic. An association with amoxicillin is possible. We added a corresponding passage in the discussion section.
Reviewer 2 Report
The paper by Pia Reinecke et al. addresses an exciting subject. The authors report their data from a prospective study of patients diagnosed with PJI. I express my sincere gratitude for the opportunity to review your manuscript. The effort of the authors is appreciated.
I would recommend addressing the following issues during the revision of the manuscript:
- “Patients received antibiotic treatment for a median of 12 weeks (range, 4 - 168 weeks)” During the discussions, the reasoning behind extending the treatment period to 168 weeks should be reported.
- “The most commonly used antibiotic was rifampin (68%, n=54), followed by sulfamethox-azole/trimethoprim (cotrimoxazole) (53%, n=42)” as rifampicin is not recommended as monotherapy, I think the antibiotics associations that were used should be discussed.
- Tabel no.1 asymptomatic bacteriuria is included as an AE attributable to antibiotic therapy. I personally do not think it is a direct relationship, you have a group of patients who, due to age and comorbidities, may have asymptomatic bacteria. My advice is to reconsider reporting this as a AE. The same can be found in table 2 - urinary tract infection and asymptomatic bacteriuria that cannot be attributed to antibiotic therapy.
- Tabel no.3 - Rifampicin is only used in combination with antistaphylococcal antibiotics. It has been most frequently associated with levofloxacin, but also with ciprofloxacin. These broad-spectrum antibiotics have additional risks of causing diarrhea or infection with Clostridioides difficile. If you have had few instances where you have identified CDI, it is important to know how can you directly link rifampicin in these cases.
- Including information about patient comorbidities would provide valuable context and help explain any potential susceptibility to adverse events, maybe can be taken into consideration.
- After assessing the data, as a personal opinion, I believe that there is one element that is inconsistent with AE. In my opinion, urinary infections should be excluded from the tables and statistics.
Author Response
Point-by-point reply
Many thanks for reviewing and commenting on our manuscript which considerably improved our manuscript.
Comments of reviewer 2:
- “Patients received antibiotic treatment for a median of 12 weeks (range, 4 - 168 weeks)” During the discussions, the reasoning behind extending the treatment period to 168 weeks should be reported.
Authors’ reply: We added the following sentences: “The antimicrobial treatment duration ranged from 4 to 168 weeks in our study. The treatment duration of several years is explained by suppressive antimicrobial treatment administered in patients suffering from infections caused by microorganisms, for which no biofilm-active agent was available or in patients, where the performed treatment strategy was not appropriate to eradicate the infection (e.g., retention of the prosthesis in case of chronic infection).”
- “The most commonly used antibiotic was rifampin (68%, n=54), followed by sulfamethox-azole/trimethoprim (cotrimoxazole) (53%, n=42)” as rifampicin is not recommended as monotherapy, I think the antibiotics associations that were used should be discussed.
Authors’ reply: Indeed, rifampin is in combination with another antimicrobial agent in order to avoid development of resistance. The administered combinations are listed in table 3 and are in line with currents recommendations.
- Tabel no.1 asymptomatic bacteriuria is included as an AE attributable to antibiotic therapy. I personally do not think it is a direct relationship, you have a group of patients who, due to age and comorbidities, may have asymptomatic bacteria. My advice is to reconsider reporting this as a AE. The same can be found in table 2 - urinary tract infection and asymptomatic bacteriuria that cannot be attributed to antibiotic therapy.
Authors’ reply: We completely agree, this is not associated with antimicrobial treatment. We excluded “urinary tract infection” and “asymptomatic bacteriuria” from the tables and statistics. Accordingly, we also corrected the total number of AEs.
- Tabel no.3 - Rifampicin is only used in combination with antistaphylococcal antibiotics. It has been most frequently associated with levofloxacin, but also with ciprofloxacin. These broad-spectrum antibiotics have additional risks of causing diarrhea or infection with Clostridioides difficile. If you have had few instances where you have identified CDI, it is important to know how can you directly link rifampicin in these cases.
Authors’ reply: In our collective, only 6 patients suffered from CDI. The low incidence despite the long administration of antibiotics may be due to the protective effect of rifampin, which has an activity against Clostridia. We added this observation/possible explanation in the discussion.
- Including information about patient comorbidities would provide valuable context and help explain any potential susceptibility to adverse events, maybe can be taken into consideration.
Authors’ reply: Unfortunately, we cannot provide this information, but as we mentioned in the discussion, as a centre we treat patients with several comorbidities. We have emphasised this again in the discussion and in the limitations.
- After assessing the data, as a personal opinion, I believe that there is one element that is inconsistent with AE. In my opinion, urinary infections should be excluded from the tables and statistics.
Authors’ reply: We completely agree, this is not associated with antimicrobial treatment. We excluded “urinary tract infection” and “asymptomatic bacteriuria” from the tables and statistics. Accordingly, we also corrected the total number of AEs.
Yours sincerely
The corresponding author, on behalf of all authors.